# The Role of Adipokines and Myokines in the Pathogenesis of Different Obesity Phenotypes—New Perspectives

**DOI:** 10.3390/antiox12122046

**Published:** 2023-11-26

**Authors:** Marta Pelczyńska, Ewa Miller-Kasprzak, Marcin Piątkowski, Roksana Mazurek, Mateusz Klause, Anna Suchecka, Magdalena Bucoń, Paweł Bogdański

**Affiliations:** 1Chair and Department of Treatment of Obesity, Metabolic Disorders and Clinical Dietetics, Poznan University of Medical Sciences, 84 Szamarzewskiego Street, 60-569 Poznań, Poland; emilka@ump.edu.pl (E.M.-K.); pbogdanski@ump.edu.pl (P.B.); 2Faculty of Medicine, Poznan University of Medical Sciences, 70 Bukowska Street, 60-812 Poznań, Poland

**Keywords:** obesity, obesity phenotypes, adipose tissue, adipokines, myokines, oxidative stress

## Abstract

Obesity is a characteristic disease of the twenty-first century that is affecting an increasing percentage of society. Obesity expresses itself in different phenotypes: normal-weight obesity (NWO), metabolically obese normal-weight (MONW), metabolically healthy obesity (MHO), and metabolically unhealthy obesity (MUO). A range of pathophysiological mechanisms underlie the occurrence of obesity, including inflammation, oxidative stress, adipokine secretion, and other processes related to the pathophysiology of adipose tissue (AT). Body mass index (BMI) is the key indicator in the diagnosis of obesity; however, in the case of the NWO and MONW phenotypes, the metabolic disturbances are present despite BMI being within the normal range. On the other hand, MHO subjects with elevated BMI values do not present metabolic abnormalities. The MUO phenotype involves both a high BMI value and an abnormal metabolic profile. In this regard, attention has been focused on the variety of molecules produced by AT and their role in the development of obesity. Nesfatin-1, neuregulin 4, myonectin, irisin, and brain-derived neurotrophic factor (BDNF) all seem to have protective effects against obesity. The primary mechanism underlying the action of nesfatin-1 involves an increase in insulin sensitivity and reduced food intake. Neuregulin 4 sup-presses lipogenesis, decreases lipid accumulation, and reduces chronic low-grade inflammation. Myonectin lowers the amount of fatty acids in the bloodstream by increasing their absorption in the liver and AT. Irisin stimulates the browning of white adipose tissue (WAT) and consequently in-creases energy expenditure, additionally regulating glucose metabolism. Another molecule, BDNF, has anorexigenic effects. Decorin protects against the development of hyperglycemia, but may also contribute to proinflammatory processes. Similar effects are shown in the case of visfatin and chemerin, which may predispose to obesity. Visfatin increases adipogenesis, causes cholesterol accumulation in macrophages, and contributes to the development of glucose intolerance. Chemerin induces angiogenesis, which promotes the expansion of AT. This review aims to discuss the role of adipokines and myokines in the pathogenesis of the different obesity phenotypes.

## 1. Introduction

Obesity is a persistent disease that constitutes a global problem. The 2017 Global Nutrition Report indicated that two billion adults are overweight [1]. According to the World Health Organization (WHO), obesity increases with the standard of living [2]. The prevalence of obesity has risen from 10% to 40% in most European countries in the last decade [3]. It has been estimated that more people are obese and overweight than are undernourished in the world. Obesity does not only involve an excess of adipose tissue (AT) in the body but is also concerned with other metabolic outcomes attributed to this tissue [4].

Adipokines and myokines are cytokines secreted by the AT and skeletal muscles, respectively. These molecules act through autocrine, paracrine, and endocrine pathways and are involved in the maintenance of energy homeostasis and the regulation of insulin sensitivity. Moreover, adipokines and myokines often function together in the human body and positively or negatively affect metabolic processes [5,6,7,8]. 

The AT is considered an active endocrine organ. It is a connective tissue composed of preadipocytes, adipocytes, stromal cells, fibroblasts, and macrophages. AT occurs in white (WAT), brown (BAT), beige, and pink tissue forms. WAT secretes a number of adipokines that affect the hunger and satiety centers, as well as metabolism processes, while BAT is involved in thermogenesis. Moreover, regional and functional depots of WAT may belong to either the visceral (VAT) or subcutaneous (SAT) adipose tissue [4,9]. VAT lines internal organs and is associated with an adverse cardiometabolic profile. The development of VAT during obesity leads to increasing levels of classically activated proinflammatory M1 macrophages. Consequently, WAT increases the amount of proinflammatory adipokines (such as visfatin and chemerin) and decreases the concentration of anti-inflammatory adipokines (nesfatin-1, neuregulin 4), contributing to a low-grade inflammatory state and the progression of obesity-linked metabolic disorders like impaired immune response and greater risk of infectious diseases [8,10,11,12]. The composition and distribution of WAT also seems to be central to the assessment of obesity phenotypes, as people in the same body mass index (BMI) category can have different metabolic risks. It has been shown that obesity phenotype can present as normal-weight obesity (NWO), metabolically obese normal-weight (MONW), metabolically healthy obesity (MHO), or metabolically unhealthy obesity (MUO). In the NWO and MONW phenotypes, metabolic disturbances are present even though the BMI value is within the normal range. In MHO, on the other hand, there is an elevated BMI value but no metabolic abnormalities. MUO phenotypes involve both high BMI values and unhealthy metabolic profiles [13,14,15]. Additionally, the generation of reactive oxygen species (ROS) results in oxidative stress, which closely affects metabolic health. Lejawa et al. noted that, in obese men, oxidative stress markers revealed a significant relationship with metabolic health in obesity phenotypes [16].

Skeletal muscle is the most abundant tissue in the human body and is responsible for generating movement and maintaining the balance of the body in normal-weight individuals. In addition, it secretes myokines and thus exerts autocrine, paracrine, and endocrine effects that lead to integrative responses in the body [17,18]. Insufficient levels of activity can lead to chronic inflammation as a result of the inappropriate release of signaling molecules and stimulation of inflammatory cells, such as neutrophils [19]. The metabolic action of myokines depends on lifestyle and on the range of physical exercises. An active lifestyle leads to favorable effects, including increased insulin sensitivity and a reduction in AT mass. Research has shown that myokines play an important role in the prevention and treatment of metabolic disorders. Physical inactivity increases the risk of metabolic diseases as it leads to the release of specific muscle-derived cytokines, such as myonectin, irisin, brain-derived neurotrophic factor (BDNF), and decorin [7,10,20].

In recent years, particular attention has been paid to a range of factors that may be involved in the pathogenesis and phenotypic presentation of obesity. These factors may include adipokines and myokines, on account of their involvement in inflammatory processes, glucose and lipid metabolism, modulation of hunger and satiety, and energy expenditure [4,15,21]. Leptin, adiponectin, and resistin are among the best-described molecules of this type [4,22]. However, studies continue to identify new adipokines and myokines, such as nesfatin-1, neuregulin 4, myonectin, irisin, BDNF, decorin, visfatin, and chemerin, which may participate in the development of obesity-related diseases. Moreover, both adipokines and myokines play a putative role in the pathogenesis of different obesity phenotypes, but their exact mechanisms of action are still not fully understood. 

This work provides a narrative review and discussion of recent data concerning the role of novel adipokines and myokines in the pathogenesis of different obesity phenotypes.

## 2. Phenotypes of Obesity

The term “obesity” is used to describe excess amounts of body AT with high bio-logical activity, although it has been noted that the composition and distribution of AT may be more relevant for the maintenance of metabolic health. Obesity mainly results from discrepancies in energy balance, which is from caloric intake being higher than caloric expenditure, but its etiology is more complex and includes environment and personal behavior, genotype–phenotype interactions, socioeconomic status, and comorbidities. Another difficulty is met in correctly diagnosing the obesity phenotype, especially in subjects with normal range BMI value [23,24,25,26].

Obesity can be diagnosed using a number of different methods, such as anthropometry, bioelectrical impedance analysis, densitometry, and imaging-based techniques, this latter including computed tomography and magnetic resonance imaging. The most commonly used methods involve the assessment of body measurements, such as weight and height, waist and hip circumference, and thickness of skin and fat folds, as well as the calculation of basic anthropometric indicators such as BMI, waist-to-hip ratio (WHR) and waist-to-height ratio (WHtR) [27]. In diagnostic, epidemiological, and screening analyses, the BMI and WHR values positively correlate with the amount of AT. On the other hand, BMI alone does not reflect the AT distribution in the body. In recent years, different obesity phenotypes have been described on the basis of body AT levels. For instance, there are people of normal weight with increased levels of body adipose, who thus have an increased risk of developing cardiovascular disease (CVD), and there are also obese individuals who demonstrate no adverse metabolic effects [25,28].

Ruderman first described metabolically obese normal-weight subjects in the 1980s. Four obesity phenotypes were later differentiated: NWO, MONW, MHO, and MUO individuals [13] (Table 1).

NWO involves a BMI within the normal weight range (<25 kg/m^2^) but a high body AT percentage. Increased VAT and decreased SAT are common in this phenotype. Despite the absence of metabolic syndrome (MetS), NWO individuals show a higher risk of metabolic disease than normal-weight individuals on account of their lower insulin sensitivity and higher blood pressure, which is accompanied by a higher concentration of inflammatory markers [29,30].

MONW is characterized by a normal BMI with a high AT percentage. An increase in both VAT and SAT accompanies MONW. Metabolic abnormalities such as insulin resistance (IR), elevated blood pressure, and increased inflammatory markers are responsible for the prevalence of MetS [24,29].

MHO involves high BMI and high body AT percentage. In this case, there are none of the metabolic abnormalities typically associated with obesity, such as high blood pressure, dyslipidemia, or IR. People with this phenotype may have normal lipid profiles and blood pressure. MHO may transform into MUO over time, so there is a higher risk of developing metabolic diseases than in individuals with normal weight [25,31].

MUO is described as a condition where BMI is above 30 kg/m^2^ and the AT percentage exceeds 30%. It also involves higher levels of VAT and lower level of SAT than in MHO. Metabolic syndrome occurs and there are accompanying conditions, such as hypertension and IR. The lipid profile is also abnormal [29,32,33,34].

It is worth adding that these criteria do not cover overweight people (BMI range between 25 and 29.9 kg/m^2^), although in clinical studies, this category is often considered as a form of obesity [35,36]. The meta-analysis of Kramer et al. demonstrated the heterogeneity of metabolic status among subjects within the same range of BMI. Overall, people with excessive body weight, even if metabolically healthy, have a greater risk of death and cardiovascular events in the long term than metabolically healthy normal-weight individuals. Similarly, metabolically unhealthy individuals (whether of normal weight, overweight, or obese) show an elevated risk of health complications [37]. It seems that in determining the risk of CVD and overall mortality, the distribution of body AT and other clinical parameters should be taken into account alongside the BMI value.

## 3. Adipokines and Myokines Protecting from Obesity

### 3.1. Nestatin-1

Nesfatin-1 is an anorexigenic neuropeptide derived from nucleobindin-2 (NUCB2) precursor, which is engaged in food intake, modulation of hunger and satiety, glucose and lipid metabolism, affective disorders, energy expenditure, and stress response (Table 2). Many studies have indicated that nesfatin-1 is secreted in both central and peripheral regions of the body and that its levels are modulated by hunger and refeeding [38,39]. Studies on rats have shown that NUCB2/nesfatin-1 is profusely expressed in several regions of the hypothalamus that play key roles in controlling food intake. These include the paraventricular nucleus of the hypothalamus (PVN, considered an integrative center for the regulation of feeding), the supraoptic nucleus, the arcuate nucleus, the lateral hypothalamic nucleus (which contains neurons expressing melanin-concentrating hormone and orexin neurons), the ventromedial hypothalamic nucleus (which triggers a counterregulatory hormone response to falling glucose levels) [40] and the solitary tract nucleus [41]. Moreover, neurons expressing nesfatin-1 colocalize with several brain transmitters. In a study by Foo et al., immunoreactivity was detected in forebrain and hindbrain nuclei, which are also immunoreactive for several peptide transmitters responsible for regulating food intake. Among these were pro-opiomelanocortin and cocaine- and amphetamine-regulated transcript (POMC/CART), α-melanocyte-stimulating hormone (α-MSH), melanin-concentrating hormone (MCH), oxytocin, somatostatin, and corticotropin-releasing hormone (CRF) [42].

It has been shown that nesfatin-1 can cross the blood–brain barrier in both directions [43], and it is, therefore, believed that peripheral nesfatin-1 has the ability to enter the brain, where it modulates appetite and nutritional reactions. The study of Zhang et al. revealed that the level of nesfatin-1 in the blood is positively correlated with BMI, IR, fasting blood glucose levels, and insulin levels [44]. The origin of the majority of circulating nesfatin-1 remains unknown. Studies of the pharmacomodulation of its signaling and the regulation of production in the periphery are still ongoing. However, the main sources of peripheral nesfatin-1 seem to be the gastrointestinal tract and the AT, which plays a role in transmitting anorexigenic signals from the periphery through adipokines, to the brain [45], affecting the integration of metabolic activity and energy balance.

The different sites of action of nesfatin-1 in the forebrain and hindbrain are emphasized by the differential onset of food intake reduction after its intracerebral injection. Studies on rodents have shown that infusion of nesfatin-1 into the third ventricle lowers meal intake among animals from one to six hours after injection. Similarly, injection of nesfatin-1 into the lateral ventricle of the rat brain decreased food intake in the dark phase with delayed start and long-term effects [46]. On the other hand, injection of nesfatin-1 directly into the PVN decreased food consumption in male rats in the three hours following the injection [47]. When injected into the fourth ventricle (4v) or cisterna magna (ic), nesfatin-1 causes a rapid and long-lasting decrease in food intake [46]. Furthermore, continuous central infusion of nesfatin-1 into the brain’s third ventricle reduces body weight increase in rats, whereas antisense oligonucleotide for NUCB2 led to weight gain after administration, confirming a physiological role for brain nesfatin-1/NUCB2 as an inhibitory body weight regulator.

The fact that the central food intake reduction effect of nesfatin-1 is sustained in leptin receptor-deficient Zucker rats suggests that its activity is independent of central leptin signaling. Leptin injection into the third ventricle (3v), in fact, does not alter the inhibitory effect of 3v nesfatin-1 on dark-phase food consumption [41]. This finding may suggest a worthwhile target for the management of obese patients, who commonly develop decreased sensitivity or resistance to leptin.

The possible relationship between nesfatin-1 and BMI has not yet been fully elucidated, with studies showing conflicting results, whether positive [44,48,49,50], negative [51,52] or the lack of correlation [53]. Also, the precise involvement of the concentration of nesfatin-1 in oxidative stress remains unclear [54]. The study of Dokumacioglu et al. showed lower nesfatin-1 and superoxide dismutase (SOD) concentrations in obese children than in the control group [55]. On the contrary, the study of Gajewska et al. found that nesfatin-1 levels positively correlated with the oxidative stress index in non-obese children with Prader–Willi syndrome, a result that may support the not fully elucidated role of nesfatin-1 in maintaining redox state [54].

Protein extracts from omental VAT of obese and average-weight patients have been analyzed to determine whether the AT of humans indeed produces nesfatin-1 and whether this protein is more abundant in obese than in non-obese individuals [56]. It was found that nesfatin-1 was undetectable in the VAT of normal-weight participants, but it was present in significant amounts in all the obese patients; this suggests that adipokine levels increase with obesity and are modulated by feeding and starvation, supporting the theory that nesfatin-1 is synthesized differentially in AT depending on the patient’s BMI [56]. While the percentage of circulating nesfatin-1 in the blood-stream did not differ by BMI or accompanying eating disorders, there were inverse associations with binge eating, emotional eating, and sweet eating (an eating behavior in which simple carbohydrates constitute at least 50% of the carbohydrates consumed per day). Moreover, positive correlations have been found with social eating and hyperphagia (a feeling of extreme, insatiable hunger). Finally, using BMI, age, gender, childhood obesity, score on the Beck Depression Inventory (BDI-II, a 21-item self-rated scale that evaluates key symptoms of depression), and score on the Binge Eating Scale (BES) as independent variables, a series of linear regression analyses were conducted to determine the characteristics associated with nesfatin-1. As an outcome, increased nesfatin-1 plasma levels were found to be linked with childhood obesity and lower BES scale scores [56].

Wang et al. investigated the effect of nesfatin-1 on the differentiation of WAT into BAT, which is a type of fat cell that burns energy to generate heat and which can thus be beneficial for weight loss and glucose homeostasis. That study found that nesfatin-1 increased the expression of genes associated with the brown adipocyte phenotype, including *Ucp1* and *Pgc-1α*, in both in vitro and in vivo models [57]. Nesfatin-1 also increased the number of multilocular lipid droplets (a characteristic trait of BAT) in white adipocytes. Furthermore, nesfatin-1 improved insulin sensitivity and glucose tolerance in mice. Overall, the findings suggest that nesfatin-1 may have therapeutic potential for the treatment of obesity and type-2 diabetes (T2D) by promoting the conversion of white adipocytes into brown adipocytes [57]. Nesfatin-1 has been studied for its effects on glucose and lipid metabolism, which are key factors in the development of obesity and metabolic disorders. It has been suggested that nesfatin-1 can improve insulin sensitivity and glucose uptake in muscle and AT while reducing hepatic glucose production and lipid accumulation, which can help prevent the development of IR and metabolic disorders, such as T2D (Figure 1) [58].

As shown in the articles and studies referenced above, nesfatin-1 has emerged as a potential regulator of energy balance, food intake, and glucose metabolism, making it a promising target for the treatment of metabolic disorders [38,58]. However, levels of nesfatin-1 may be affected in different types of obesity phenotypes, which may affect its potential therapeutic value. It is, therefore, essential to look into the numerous pathways in which nesfatin-1 levels may be altered in different obesity phenotypes to shed light on the complex nature of this hormone’s role in metabolism and its therapeutic potential. 

Studies indicate that MHO-like individuals demonstrate higher levels of nesfatin-1 than MUO-like subjects. In the of Başar et al. [59], obese people were found to have considerably lower serum nesfatin-1 concentrations than non-obese individuals (Table 2). Another study [51] that analyzed the association between obesity and nesfatin-1 had similar results. Moreover, BMI, body AT percentage, body AT weight, and blood glucose levels were all found to be negatively correlated with nesfatin-1 levels [51]. Those differences may result from the fact that individuals with MUO-like phenotype have more severe metabolic dysfunction, which may have an impact on nesfatin-1 production and function. In a comparable manner, Alotibi et al. found that nesfatin-1 levels were lower in obese people with MetS than in those who did not have metabolic outcomes [60]. They also discovered a link between nesfatin-1 levels and MetS components like BMI, waist circumference, and body weight [60].

Overall, while the role of nesfatin-1 in different phenotypes of obesity is not fully understood, it appears that nesfatin-1 levels may be altered in response to changes in body weight, AT distribution, and metabolic function. Nesfatin-1 may thus be involved in the maintenance of MHO.

**Table 2 antioxidants-12-02046-t002:** The potential role of adipokines and myokines in obesity and its phenotypes.

Adipokines/Myokines	The Potential Role in Obesity and Its Phenotypes	References
Nesfatin-1	Regulator of energy homeostasis, food intake, and glucose metabolism; potential anorexigenic factor;	[38]
	Obese people (especially MUO-like phenotype) demonstrate lower nesfatin-1 concentrations than non-obese individuals;	[51,59]
	Nesfatin-1 levels negatively correlate with BMI, body AT percentage, body AT weight, and blood glucose levels.	[51]
Neuregulin 4	Inhibits lipogenesis and lipid accumulation, reduces chronic inflammation;	[61]
	Neuregulin 4 is decreased in obese individuals with metabolic syndrome (MUO-like phenotype) and negatively correlated with waist circumference, body AT percentage, BMI, LDL cholesterol, and fasting glucose concentration.	[62]
Myonectin	Influences lipid homeostasis in the liver and AT, regulates energy metabolism;	[63]
	Myonectin can be correlated to MHO-like phenotype; exercises increase levels of myonectin in obese subjects and decrease IR.	[64]
Irisin	Supports WAT browning, increases energy expenditure, regulates glucose metabolic homeostasis;	[65]
	Reduces inflammatory processes;	[66,67]
	Conflicting results occur in irisin concentration in obesity phenotypes. The decrease levels of irisine in MUO obese has been showed.	[68,69]
	While, another study indicated its increase concentration.	[70]
Decorin	Is involved in inflammatory processes and maintaining glucose tolerance;	[71]
	Decorin concentration is increased in obesity.	[72]
BDNF	Possess anorexigenic effects; regulates energy homeostasis;	[73,74]
	Conflicting results occurs BDNF concentration in obesity. Data indicated its decrease concentration.	[75]
	On the other hand, another study indicated its increase levels.	[76]
Visfatin	Enhances adipogenesis, promotes pro-inflammatory processes and IR, contribute to cholesterol accumulation;	[77,78]
	Obese individuals (MUO-like phenotype) present higher visfatin levels;	[79]
	Visfatin levels positively correlate with IR indicators such as glucose and insulin concentration and HOMA-IR index value.	[80]
Chemerin	Promotes AT growth by inducing angiogenesis and increasing its vascularization, increases inflammation in AT;	[81]
	Chemerin levels are lower in MHO compared to MUO;	[82]
	In patients with obesity, chemerin levels positively correlates with obesity markers (HOMA-IR, BMI, AT percentage, waist circumference, WHR, triglycerides, total cholesterol).	[83]

Abbreviations: AT, adipose tissue; BDNF, brain-derived neurotrophic factor; BMI, body mass in-dex; IR, insulin resistance; HOMA-IR, homeostatic model assessment-insulin resistance; MHO, metabolically healthy obese; MUO, metabolically unhealthy obese; WHR, waist to hip ratio.

### 3.2. Neuregulin 4

Neuregulin 4 (Nrg4) belongs to the neuregulin family of extracellular ligands that activate type-1 growth factor receptors (ErbB-3 and ErbB-4 receptors). Nrg4 primarily activates ERbB-4 and transmits signals across cell membranes through tyrosine phosphorylation [84]. This factor is enriched in BAT and has been detected in the pancreas, muscles, breast milk, developing intestine tissue, and in prostate, breast, and gastric cancers [85,86,87]. 

Nrg4 dampens hepatic lipogenic signaling and maintains glucose and lipid homeostasis in the context of obesity. These results come out of the studies conducted by Wang et al. [88], which investigated the reduced expression of Nrg4 in obese rodents on a high-fat diet (HFD) and humans. The findings of these studies indicate that, upon examining the mRNA levels of *Nrg4* in the SAT of obese individuals, body AT percentage and hepatic fat content is found to be inversely correlated. Lower levels of *Nrg4* mRNA were observed in participants who exhibited impaired glucose tolerance or T2D than in individuals with normal glucose tolerance. The research of Ma et al. [61] indicated that gene transfer of *Nrg4* to HFD-induced obese mice prevents metabolic changes associated with obesity (Table 2). These results reveal that Nrg4 attenuates weight gain; it additionally reduces chronic inflammation and IR, and prevents hepatic steatosis by inhibiting lipogenesis and lipid accumulation (Figure 1). Overexpression of the *Nrg4* gene results in the decreased expression of monocyte chemotactic protein 1 (MCP-1), a chemokine gene involved in the regulation of macrophage migration and infiltration. Conversely, *Nrg4* gene transfer increased the expression of the M2 macrophage marker CD163. These findings indicate that Nrg4 has anti-inflammatory effects when overexpressed, mitigating diet-induced obesity [61]. Nrg4 is believed to affect inflammation and oxidative stress states. Patients with diabetic peripheral neuropathy presented reduced circulation levels of Nrg4 than controls [89]. Similar results were seen in patients with newly diagnosed T2M with metabolic syndrome when compared to counterparts without metabolic syndrome [90].

It is worth mentioning that the study of Cai et al. [62], who examined 1212 obese adults, shows that circulating levels of Nrg4 are negatively correlated with waist circumference, body AT percentage, and BMI and LDL cholesterol concentration (Table 2). Additionally, lower levels of Nrg4 were observed in subjects with hyperglycemia and were inversely associated with fasting glucose concentration. Furthermore, data from this study demonstrated that individuals with MetS had significantly lower circulating Nrg4 levels than the control group. Based on these results, we can infer that the control group (obese adults without MetS) represents individuals with the MHO phenotype, while the group of obese participants with MetS represents individuals with the MUO phenotype. 

The above studies demonstrate that levels of Nrg4 are lower in individuals with the MUO-like obesity phenotype than in those with the MHO-like phenotype, and this may protect against metabolic disorders accompanied by excessive body weight. However, further research is needed to determine the precise correlation between Nrg4 concentration and obesity phenotype.

### 3.3. Myonectin

Myonectin, described as CTRP 15 (C1q/TNF-related protein isoform 15), is a myo-kine with obesoprotective effects that may represent a new therapeutic target. Skeletal muscles are responsible for the secretion of myonectin, a member of the CTRP family [63,91]. Myonectin is responsible for regulating the integration of the whole-body metabolism. Research indicates that myokine affects lipid homeostasis in the liver and AT, in dependence on the energy metabolism status within the cells [63,91]. 

Myonectin resembles the action of insulin by activating similar signaling pathways and increasing the expression of CD36, caveolin 1 (Cav1), and fatty acid binding proteins (Fabp1/Fabp4) in adipocytes and hepatocytes. Myonectin circulates in the blood, and its elevated levels decrease concentrations of nonesterified free fatty acid (NEFA) (Figure 1) [63]. Studies have provided evidence that this myokine is involved in the regulation of lipid metabolism through its ability to inhibit adipogenesis. This is achieved in 3T3-L1 preadipocytes through a decrease in the expression of adipogenic transcription factors such as cytosine–cytosine–adenosine–adenosine–thymidine enhancer-binding proteins (C/EBP) α and β and peroxisome proliferator-activated receptor gamma (PPARγ), as well as by blocking regulation of the p38 mitogen-activated protein kinase (MAPK) and C/EBP homologous protein (CHOP) adipogenic pathway [91]. 

The effect of myonectin on fatty acid metabolism has also been confirmed in a study on mice by Seldin et al. [63]. It has been shown that obesity, which is associated with a dysregulation of overall metabolism, reduces levels of myonectin. During fasting, the concentration of myonectin decreases, while refeeding dramatically increases its mRNA and serum levels [63]. This study provides evidence that the expression of myonectin increased under the influence of glucose and fatty acids (Table 2) [63]. Furthermore, although mRNA and circulating levels of this myokine were lower in HFD-obese mice, exercise increased its expression and concentration, which protected against IR [63]. Myonectin also reduced the concentration of NEFAs by capturing them in the liver and AT. 

Another study found that physical activity, especially aerobic training, significantly contributes to the prevention of obesity development and its complications, partially through the effects of myonectin on this process [64]. This study involved obese and overweight women with increased BMI and an MHO-like phenotype. Serum levels of myonectin were higher in the women who performed exercises than in the nonexercising women. Greater tissue sensitivity to insulin was also seen in the first group after physical exercise. Nevertheless BMI decreased in the group of exercising women [64]. The study suggests that it was aerobic physical exercise that caused the myonectin levels to increase, in turn promoting better nutrient absorption and normalizing lipid metabolism and IR in individuals with obesity and overweight. The study also demonstrated that insulin-dependent factors were decreased in obesity as a result of exercise (Table 2) [64]. The accumulation of fatty acids in the body can also be reduced through physical activity. It has been shown that muscle contraction improves insulin action by increasing the permeability of glucose through glucose transporter-4 (GLUT-4) [64]. The action of myonectin may be correlated with the MHO-like phenotype, as in the case of the women with overweight or obesity who showed increased BMI but no metabolic diseases. Nevertheless, data relating obesity phenotypes with myonectin levels are limited.

### 3.4. Irisin

Irisin was first described by Boström et al. in 2012 [92], and research on its effect on the body has continued. This myokine induces the browning of AT and has been shown to have a considerable effect on the regulation of glucose metabolism. This myokine has a number of properties, affecting the liver, bones, and AT, and promotes a normoglycemic state [65,93,94].

Irisin is a myokine formed by the cleavage of fibronectin type III domain containing 5 proteins (FNDC5) by proteins of the disintegrin and metallopeptidase (ADAM) family, such as ADAM10, to form the 112 aa peptide irisin [95,96,97]. Most likely, irisin combines with integrin αVβ1 in adipocytes [98,99].

It has been shown that WAT browning supported by irisin conduces to increased energy expenditure, regulates glucose metabolic homeostasis by promoting glycogen synthesis, and inhibits gluconeogenesis in the liver (Table 2) [65]. Beige adipocytes have been described as thermogenic cells formed from WAT that play a role in thermogenesis and energy expenditure [100,101,102]. They are similar to BAT in that they have higher levels of transcription of the uncoupling protein 1 (UPC1) gene as well as the ability to perform thermogenesis. UPC1 is located in the inner membrane of the mitochondrion. It uncouples electron transport from ATP production. The BAT content of the body decreases with aging, resulting in less thermogenic capacity from BAT, but the large amount of WAT that can be transformed into beige adipocytes would seem to contribute significantly to the ability to dissipate energy into heat. This suggests that WAT browning may reduce obesity while simultaneously reducing the risk of related complications [103].

Irisin seems to modify these processes by stimulating glycogen synthesis and inhibiting gluconeogenesis in the liver, thus regulating glucose metabolic homeostasis [104,105,106]. Moreover, irisin is presumed to inhibit the binding of myeloid differentiation factor 2 and toll-like receptor 4 (TLR4), resulting in attenuation of the inflammatory response [107]. Irisin, by affecting the metabolic regulation of glucose and cholesterol synthesis in the liver, may prevent the development of obesity (and diabetes) [108].

Irisin is mainly produced in skeletal muscle, although smaller amounts are also synthesized in AT, cardiomyocytes, and bones [109,110]. An elevated calcium concentration in the cytoplasm of skeletal muscle during contraction determines the transcription and activation of the peroxisome proliferator-activated receptor gamma 1-alpha (PGC-1α), which regulates genes involved in human metabolism. An upregulation of PGC-1α expression leads to cleavage of the FNDC5 factor, resulting in the formation of irisin [92,111]. This way, the amount of irisin increases as a result of skeletal muscle work. Resistance exercise has been found to be the best stimulator of irisin expression [112].

Individuals with an excessive body weight usually have higher levels of proinflammatory markers. Moreover, obesity causes chronic inflammation with low levels of malignancy and may also promote IR [113,114]. An excess of WAT contributes to increased gene expression of inflammatory pathways, resulting in higher levels of tumor necrosis factor α (TNF-α) and interleukin 6 (IL-6) [115]. The presumed anti-inflammatory properties of irisin have been demonstrated through its positive effect on the phagocytosis capacity of macrophages, with it modulating their activity and reducing ROS production [67]. The antidiabetic and antioxidant action of irisin was also suggested in the study of Belviranly et al., which examined premenopausal women with obesity [116]. Other authors also documented lower serum irisin concentration in obese patients with T2M than in normal-weight controls. One study noted a negative correlation between resistin and interleukin-6 in obese individuals with T2DM [117].

A study conducted by Mazur-Bialy et al. showed a reduction in the inflammatory process in cultured lipopolysaccharide-activated adipocytes through irisin’s inhibition of the formation of proinflammatory cytokines such as TNF-α and IL-6. They also demonstrated that irisin causes a decrease in leptin synthesis and an increase in adiponectin levels (Table 2) [66]. Another study indicated that the decreased secretion of IL-6 and TNF-α accompanied by increased levels of the anti-inflammatory interleukin 10 (IL-10) in VAT and SAT can be attributed to irisin [118]. Although the anti-inflammatory potential of irisin has been demonstrated [66,67,118] further studies are needed to better understand its mechanisms of action.

The relationship between irisin levels and obesity is controversial. Several reports have shown that irisin expression is lower in obese individuals and in patients with T2D than in healthy subjects [119,120,121]. However, there are studies in which higher irisin levels have been noted in obese people than in non-obese individuals [55,122,123]. Several factors may contribute to these contradictory results: it is unknown whether irisin sensitivity decreases in the WAT or whether the irisin receptor becomes nonfunctional [124,125]. In spite of these doubts, a great number of reports have pointed to a relationship between irisin levels and certain markers of adiposity, which suggests that the concentration of this myokine may reflect net body adiposity.

It is worth adding that some studies have tested irisin levels in different obesity phenotypes. Yosaee et al. observed that irisin levels in MHO and MUO were lower than in the control group of nonobese metabolically healthy participants. Further, its concentration was lowest in individuals with obesity accompanied by metabolic disorders (Table 2) [68]. These results were confirmed by Castillo et al. in MHO and MUO children compared to their normal-weight age-mates [69]. On the other hand, a study by Abulmeaty et al. showed higher irisin levels in MUO individuals than in normal weight-lean subjects. Moreover, irisin levels were also elevated in their MHO group [70]. In another study, NWO participants presented higher concentrations of irisin than the control group (Figure 1) [126]. These ambiguous results should be clarified through further studies focusing on the association of irisin with obesity phenotypes.

### 3.5. Decorin

Decorin is an extracellular matrix (ECM) protein with a multifunctional nature that affects metabolism. It belongs to the first class of the small leucine-rich proteoglycan (SLRP) family [127]. Decorin is released from myocytes into the bloodstream in response to muscle contractions. It has been shown that decorin may have significant implications for obesity and AT metabolism [128].

Expression of decorin has been observed in adipose tissue, especially in VAT [129]. In addition, decorin interacts with various other ECM molecules, such as myostatin, fibronectin, and elastin, as well as TNF-α and epidermal growth factor (EGF), which are growth factors [130]. Decorin, as a component of SLRP, participates in the transduction of cell signaling and interacts with cell proliferation and differentiation [131]. Decorin activates various receptors on the surface of macrophages [127]. It has been shown that decorin acts as a ligand and binds to toll-like receptor 2 (TLR2) and TLR4, leading to the activation of signaling pathways such as p38 mitogen-activated protein kinase (MAPK), extracellular signal-regulated kinase 1/2 (ERK1/2), and nuclear factor kappa B (NF-κB). These reactions induce the synthesis of proinflammatory cytokines, such as TNF-α and interleukin-12p70 (IL-12) (Table 2) [71]. Decorin also triggers proinflammatory factors through interactions with receptors for advanced glycation end products (AGER) on macrophages in an NF-κB dependent manner [132]. In addition, decorin inhibits the anti-inflammatory effect of IL-10 by blocking transforming growth factor-β (TGF-β) [127]. Moreover, it has been noted that decorin participates in autophagy in endothelial and epithelial cells and regulates the functioning of immune cells, which may allow us to hypothesize that it is involved in inflammatory and autoimmune diseases [127]. 

Decorin may participate in immune response and induce inflammatory processes, but it is also associated with AT homeostasis and metabolic response. Recently, Hirata et al. studied proteome and oxidative stress protection potential in diet-induced obese rats treated with *Ginko biloba* extract and found that upregulation of decorin was accompanied by downregulation of oxidative stress-related protein [133]. It has recently been observed that decorin is responsible for maintaining glucose tolerance (Figure 1) [130]. Studies showed that decorin expression was found to be increased in the AT of rodents and obese humans [129,134]. Unfortunately, no information was mentioned about obesity phenotypes. Svärd et al. indicated that the decorin knock-out C57BL/6J mice (DcnKO) showed increased feed efficiency upon overfeeding, as well as impaired glucose tolerance. Moreover, circulating leptin levels and adipose leptin mRNA were higher in DcnKO mice. They concluded that decorin may play an important role by protecting against diet-induced hyperglycemia [130]. Bolton et al. confirmed these results, showing that plasma decorin levels in humans with T2D were 12% higher than in the control group (with normal glucose concentration). Furthermore, they showed a significant correlation between decorin concentration and WHR value [129]. Obesity is often accompanied by low-grade inflammation, and decorin has been shown to act as a proinflammatory molecule. However, decorin probably binds the proinflammatory activity of the complement protein C1q42 and antagonizes C1q-induced IR [135]. These hypotheses indicate that decorin may play a role in inflammation response in AT, though the relationship between it is proinflammatory and anti-inflammatory activities is complex. 

Another study has demonstrated that decorin may reduce the number of adipocytes through the activation of caspase-3, a protein involved in apoptosis [136]. This study showed that decorin regulates glucose levels, primarily affecting glucose metabolism by protecting the body from potential postprandial hyperglycemia. Interestingly, despite the impaired glucose tolerance in the DcnKO mice, no evidence of increased adipocyte size was found. This suggests that the absence of decorin (independently of adipocyte hypertrophy) negatively affects glucose homeostasis [130].

Based on these results, it is difficult to attribute a specific role to decorin in obesity, although it has been observed that its concentration may be increased in obese subjects, as compared with non-obese people [72] and with individuals with impaired glucose tolerance [129]. Unfortunately, no association has been described between decorin and obesity phenotypes. Studies have demonstrated the protective effects of decorin on glucose metabolism, which may suggest it has a role in reducing metabolic disorders correlated with an excessive amount of AT. However, decorin may also promote an inflammatory response, which suggests it has a dual nature.

### 3.6. BDNF

The amount of food consumed is an important factor in the development of obesity. Various disorders in the stimulation of the nervous system that lead to increased hunger can cause excessive weight gain. Recently, considerable attention has been paid to BDNF, with studies attributing this myokine with anorexigenic effects (Figure 1) [73]. 

BDNF is a neurotrophic growth factor [137] that is also referred to as a myokine [138]. It contributes to the regulation of energy homeostasis in the human body (Table 2) [74], plays a significant role in synaptic plasticity, neurogenesis, and neuronal protection, and is believed to be involved in the neurophysiological mechanisms of appetite control, demonstrating anorexigenic properties [139].

BDNF acts via tropomyosin-related kinase B (TrkB) receptor [140]. The BDNF-TrkB pathway interacts with intracellular pathways, including the neuromodulatory processes of gamma-aminobutyric acid and dopamine [141]. The BDNF-TrkB pathway has been shown to be important in the body’s energy homeostasis [140].

As in the case of decorin above, there are, unfortunately, no studies relating obesity phenotypes to BDNF action. Nevertheless, studies have been made to determine BDNF levels in patients with a BMI of 30 kg/m^2^ or more. In studies of obese patients and those of normal weight, BDNF has been shown to be differentially related to brain responses to food cues. Patients with obesity reported a significant positive correlation between visual food cue-reactivity and plasma BDNF levels. The same study, however, did not observe different BDNF levels or differences in food cravings in obese participants compared to normal-weight participants [142]. It can be inferred that a defective BDNF molecule or a defective TrkB receptor contributes to obesity through impaired control of food intake but does not directly result in stimulation of AT formation [143,144]. This is due to a reduced anorexigenic effect. It has been demonstrated that a mutation of the *NTRK2* gene resulting in a non-functioning TrkB receptor contributes to obesity through hyperphagia [145].

The anorexigenic properties of BDNF have attracted attention aimed at better understanding its impact on the pathogenesis of eating disorders and its possible use in their treatment. It has been concluded that peripheral BDNF levels are lower in obese children than in normal-weight subjects (Table 2) [75], but other meta-analyses have not confirmed this thesis, showing no differences between the obese and control groups [146]. Moreover, there have been cases in which higher levels of BDNF were observed in obese people [76]. Elsner et al. studied LPS-treated peripheral blood mononuclear cells (PBMC) from obese individuals who performed exercise and from an obese sedentary group and found that the physically active obese patients had increased levels of BDNF and SOD activity compared to their sedentary counterparts [147]. The many hypotheses and contradictory results do not currently allow a clear definition of the role of BDNF in the development of metabolic disorders and obesity.

## 4. Adipokines and Myokines Predisposing to Obesity

### 4.1. Visfatin

Visfatin, also known as pre-B cell colony-enhancing factor (PBEF) or nicotinamide phosphoribosyltransferase, is a protein that was first identified as a pre-B cell growth factor. Visfatin also acts as an important factor involved in energy metabolism and inflammatory processes. It seems that visfatin expression is modulated by cytokines that increase IR, such as TNF-α, IL-1β, and IL-6. Moreover, visfatin has biological properties comparable to those of growth factors, including anti-apoptotic properties and the ability to promote cell proliferation [77,148,149].

Visfatin is expressed in numerous tissues, including AT (especially visceral), muscles, liver, and in cells of the immune system. It alters glucose and lipid metabolism and has insulin-mimetic effects [150]. Visfatin has been confirmed to interfere with insulin signaling pathways, resulting in lower glucose absorption in adipocytes along with other tissues susceptible to insulin. Moreover, it has been indicated that visfatin can enhance adipogenesis through the promotion of new AT cells, and consequently leads to the development of obesity-induced IR (Figure 1) [77].

Visfatin may also regulate lipid metabolism. It has been suggested that visfatin may indirectly contribute to cholesterol accumulation by modulating the expression of scavenger receptors (SR)-A and CD36 in macrophages during the formation of foam cells [151]. Visfatin can induce the expression of proinflammatory cytokines such as TNF-α and IL-8 in macrophages. The potential pro-atherogenic effects of visfatin increased the risk of cardiovascular events, especially in patients with abdominal obesity [78]. Visfatin promotes vascular inflammation, proinflammatory cytokine and chemokine secretion, macrophage survival, vascular smooth muscle inflammation, leukocyte recruitment of endothelial cells, and, in consequence, plaque destabilization. Thus, in circulatory system diseases—and particularly with atherosclerosis, endothelial dysfunction, and vascular damage—visfatin has been proposed as a useful prognostic tool [152]. Interestingly, the count of circulating progenitor endothelial cells deter-mined from obese patients has been found to be decreased while serum visfatin and oxidative stress markers increased, suggesting an interaction between visfatin concentration, endothelial function, and obesity [153].

Visfatin plasma concentration particularly corresponds with intra-abdominal (visceral) AT mass. Additionally, the fact that visfatin levels rise after an HFD shows that visfatin has an effect on obesity-induced IR [77,154]. Several studies have reported that circulating visfatin levels are high in subjects with metabolic diseases, such as obesity and T2D. Moreover, visfatin concentration is often positively associated with IR [155,156,157] and inflammatory markers in individuals with obesity, particularly those who have co-occurring metabolic abnormalities (MUO-like phenotype). According to Lejawa et al., MUO subjects had higher levels of visfatin than did people with MHO and MHNW, implying that this adipokine may have an effect on the development of obesity-related metabolic disorders (Table 2) [79].

Sethi and Vidal-Puig suggested that visfatin may be the missing link between intra-abdominal obesity and diabetes. Their article reviews several studies that have found elevated levels of visfatin in obese individuals, particularly in those with intra-abdominal obesity, and they suggest that this may contribute to the development of IR and T2DM. Their study indicates that the production of visfatin is low in the lean state and that its effect on insulin sensitivity may be insignificant. On the other hand, intra-abdominal obesity increases visfatin synthesis, which may enhance obesity while maintaining insulin sensitivity in the peripheral organs [77]. Nourbakhsh et al. measured visfatin levels in obese children with and without MetS. They discovered that children from the research group (MUO-like phenotype) had higher visfatin levels than those from the control (without MetS). What is more, visfatin levels were positively correlated with IR indicators such as glucose and insulin concentration and HOMA-IR index value (Table 2) [80]. Those results have been confirmed by other authors in both children [158,159] and adults [160].

Kim et al. [161] summarized several studies that have investigated the relationship between visfatin levels and metabolic abnormalities, including glucose intolerance, IR, and dyslipidemia. They have further suggested that visfatin may contribute to IR by promoting the activation of inflammatory pathways and impairing insulin signaling (e.g., via the STAT3 and NF-κB pathways in HepG2 cells) [154] in targeted tissues such as the liver, muscle, and AT. Visfatin may also play a role in the regulation of energy metabolism, lipid metabolism, and glucose homeostasis. It has been found that visfatin levels are positively associated with IR and markers of inflammation in individuals with MUO-like phenotype but not in those with MHO-like phenotype or in normal-weight individuals. Kim et al. suggested that visfatin may contribute to the development of metabolic complications in individuals with MUO-like phenotype and may be a potential therapeutic target for the prevention and treatment of these conditions [161].

### 4.2. Chemerin

Chemerin is a chemotactic factor and an adipokine secreted by adipocytes. It acts as a ligand for the G protein-coupled receptor (ChemR23) involved in immune responses. The chemerin gene (*RRARES2*) is expressed in WAT, as well as in the liver and lungs. The ChemR23 gene (*CMKLR1*) is expressed in dendritic cells, monocytes, macrophages, and endothelial cells [162,163,164,165] and is enhanced by proinflammatory cytokines such as TNF-α and IL-6 [166]. The expression of chemerin shows a positive correlation with the expression levels of proinflammatory cytokines [167]. Studies have demonstrated that chemerin is capable of activating critical angiogenic signaling pathways and promoting angiogenesis in in vitro experiments [166,168]. Higher levels of chemerin during adipogenesis may thus promote AT growth by inducing angiogenesis and increasing its vascularization. The chemotactic function of chemerin for immune cells may contribute to the inflammation of WAT in obesity (Figure 1) [81].

Numerous studies indicate that levels of chemerin are increased in obese adults and children [168,169,170,171]. A meta-analysis has also revealed a positive correlation between obesity markers (such as HOMA-IR, BMI, body AT percentage, waist circumference, and WHR values, as well as triglycerides, total cholesterol, and CRP concentration) and chemerin levels in patients with obesity or MetS (Table 2) [83]. Fülöp et al. documented the negative correlation between serum chemerin concentration and the paraoxonase 1 arylesterase activity associated with antioxidant capacity [172].

The study of Corona-Meraz [173] observed that CMKLR1 transcript and chemerin levels were elevated in obese individuals without IR. Additionally, chemerin concentration correlated positively with adiposity and metabolic markers (including waist and hip circumference and triglycerides). The above information suggests that chemerin concentration and CMKLR1 expression exhibit inverse expression patterns in relation to the low-grade inflammatory response associated with IR development and that higher levels of chemerin in obesity may promote dysmetabolic responses. Ernst et al. [174] studied CMKLR1 receptor deficiency in mice and found that receptor loss was associated with reduced body weight gain and obesity levels in CMKLR1−/− mice. The animals also exhibited lower fasting blood glucose levels, serum insulin levels, and decreased mRNA levels of proinflammatory cytokines. These studies, along with that of Sanidas et al. [82], which investigated chemerin levels in a hundred individuals with MHO and in a hundred individuals with MUO, may allow us to infer that chemerin levels are lower in MHO than in MUO (Table 2). However, further investigation is required to elucidate the exact association between chemerin levels and obesity phenotype.

## 5. Conclusions

Obesity is a multifaceted condition associated with a range of health issues, such as cardiovascular disease, type-2 diabetes, and metabolic syndrome. Myokines and adipokines are molecules secreted by skeletal muscle and adipose tissue (AT), one of whose roles is endocrine activity. Physical activity has a beneficial impact on the expression of myokines, increasing the production of anti-inflammatory cytokines, which inhibits the further development of obesity. It has been shown that adipokines demonstrating protective effects, such as nesfatin-1 and neuregulin 4, often occur in elevated concentrations in metabolically healthy obesity (MHO) as compared to metabolically unhealthy obesity (MUO). The connection between irisin concentrations and obesity phenotypes is unclear, and it is difficult to determine the role of myokines generally in obesity phenotypes. Research suggests that myonectin may potentially contribute to the MHO phenotype. The proinflammatory effects of the adipokines and myokines surveyed in this article, such as visfatin and chemerin, contribute to the development of obesity. People with MHO have lower concentrations of these adipokines than those with MUO. However, there are still not enough studies to determine the exact effect of these molecules on metabolism, and in particular on obesity phenotypes. Further research is thus warranted to assess the contribution of adipokine and myokine networks in obesity phenotypes, with the aim of identifying molecules of potential prognostic or diagnostic significance.

## Figures and Tables

**Figure 1 antioxidants-12-02046-f001:**
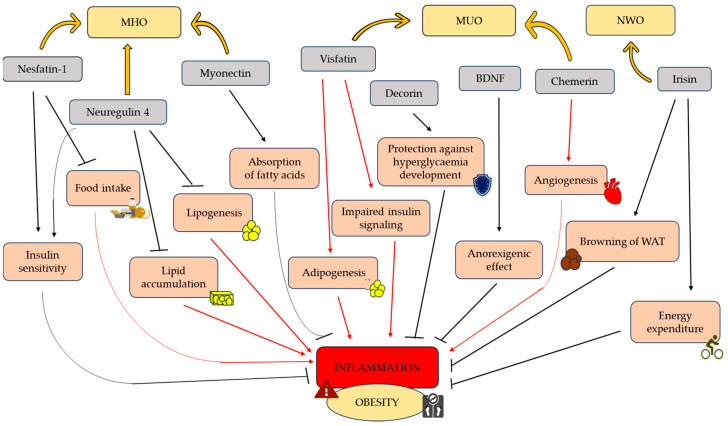
The participation of adipokines and myokines in obesity-induced inflammation. Abbreviations: BDNF, brain-derived neurotrophic factor; MHO, metabolically healthy obese; MUO, metabolically unhealthy obese; NOW, normal-weight obese; WAT, white adipose tissue. This figure was made using the Servier Medical Art collection (http://smart.servier.com/, accessed on 25 October 2023).

**Table 1 antioxidants-12-02046-t001:** The characteristics of the different phenotypes of obesity.

	NWO	MONW	MHO	MUO
BMI (kg/m^2^)	<25 (normal)	<25 (normal)	>30 (elevated)	>30 (elevated)
Body AT percentage	↑	↑	↑	↑
VAT	↑	↑	Lower compared to MUO	↑
SAT	↓	↑	Higher compared to MUO	↓
Insulin sensitivity	↓	↓	Favorable	↓
Inflammatory markers	↑	↑	Normal	↑
Blood pressure	↑	↑	Normal	↑
MetS	−	+	−	+

Abbreviations: AT, adipose tissue; BMI, body mass index; MHO, metabolically healthy obese; MUO, metabolically unhealthy obese; MONW, metabolically obese normal-weight; NOW, normal-weight obese; MetS, metabolic syndrome; SAT, subcutaneous adipose tissue; VAT, visceral adipose tissue; ↑, increase; ↓, decrease; + presence; − absence.

## Data Availability

Not applicable.

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
