# Peer review of "The Role of Adipokines and Myokines in the Pathogenesis of Different Obesity Phenotypes—New Perspectives"

_antioxidants, 2023, doi:10.3390/antiox12122046_

Round 1

Reviewer 1 Report

Comments and Suggestions for Authors

Title: The role of adipokines and myokines in the pathogenesis of different obesity phenotypes – a new perspectives

Authors: Marta Pelczyńska, Ewa Miller-Kasprzak, Marcin Piątkowski, Roksana Mazurek, Mateusz Klause, Anna Suchecka, Magdalena Bucoń, Paweł Bogdański.

General Comments:

Obesity is a common; however, heterogeneous disease that hampers its proper diagnosis and treatment. Therefore, there is a need for basic research that would help to elucidate its complex pathogenesis and hopefully improve the management of the disease. In theirs work, Marta Pelczyńska et al. discussed the role of several recently discovered adipokines and myokines in the pathogenesis of different obesity phenotypes. The concept of this review is clear, and it constitutes a valuable contribution to the field; however, there are some aspects the Authors should consider before the manuscript is accepted for publication.

Major revisions:

Whole manuscript:

1)     Please justify focusing on the novel adipokines and myokines and only mentioning the role of "classical" adipokines (leptin, adiponectin, resistin) in the pathogenesis of different obesity phenotypes

2)     Please consider adding a table summarizing the role of the described adipokines in human health and the pathogenesis of different obesity phenotypes.

3)     Please add information if the review is a narrative or a systematic one. In the second case, please add a material and methods section to clarify what selection criteria for the cited studies were used.

Section: Phenotypes of obesity

1)     The presented criteria for tdiagnosing different obesity phenotypes do not cover the group of patients with overweight (BMI 25-29.9 kg/m2). Please explain/suggest how these patients should be classified.

Section: Conclusions:

1)     The Authors write: “Understanding the underlying causes associated with various phenotypes of obesity may contribute to the development of a better therapeutic strategy, reducing the number of cases and complications and thereby enhancing the quality of life of the patients with an excessive body AT content.” – please discuss if are any examples of therapeutic strategies from pre- or clinical studies aiming at described adipokines/myokines in treatment of obesity and/or its complications.

Minor revisions:

Whole manuscript:

1)      The manuscript may benefit from the assistance of a native English speaker.

Introduction:

1)      “Skeletal muscles, as the most abundant tissue in the human body, are responsible for generating movements and maintaining the balance of the body." – please add "in the normal-weight individuals."

Comments on the Quality of English Language

The manuscript may benefit from the assistance of a native English speaker.

Author Response

RESPONSE TO REVIEWER 1:

Thank you for careful reading of our manuscript. All of the suggestion have been included in the modified version of the article and marked up using the “Track Changes” function in the text. We would like to also admit, that considering all Reviewers suggestions, our manuscript was improved in general.

Whole manuscript:

  • Comment no 1.

Please justify focusing on the novel adipokines and myokines and only mentioning the role of "classical" adipokines (leptin, adiponectin, resistin) in the pathogenesis of different obesity phenotypes.

Thank you for your comment. We added the sentence justifying the description of novel adipokines and myokines: “Leptin, adiponectin, and resistin are among the best-described molecules of this type [4,22]. However, studies continue to identify new adipokines and myokines, such as nesfatin-1, neuregulin 4, myonectin, irisin, BDNF, decorin, visfatin, and chemerin, which may participate in the development of obesity-related diseases. Moreover, both adipokines and myokines play a putative role in the pathogenesis of different obesity phenotypes, but their exact mechanisms of action are still not fully understood” (page 2 line 96).

  • Comment no 2.

Please consider adding a table summarizing the role of the described adipokines in human health and the pathogenesis of different obesity phenotypes.

Thank you for your valuable comment. The Table has been added (page 7 line 290).

  • Comment no 3.

Please add information if the review is a narrative or a systematic one. In the second case, please add a material and methods section to clarify what selection criteria for the cited studies were used.

According to the Reviewer suggestion, the information has been added. The review is a narrative one (page 3 line 102).

Phenotypes of obesity:

  • Comment no 1.

The presented criteria for diagnosing different obesity phenotypes do not cover the group of patients with overweight (BMI 25-29.9 kg/m2). Please explain/suggest how these patients should be classified.

Thank you for your careful reading and comments. We added the paragraph explaining above mentioned concerns: “It is worth adding that these criteria do not cover overweight people (BMI range between 25 and 29.9 kg/m2), although in clinical studies this category is often consid-ered as a form of obesity [35,36]. The meta-analysis of Kramer et al. demonstrated the heterogeneity of metabolic status among subjects within the same range of BMI. Over-all, people with excessive body weight, even if metabolically healthy, have a greater risk of death and cardiovascular events in the long term than metabolically healthy normal-weight individuals. Similarly, metabolically unhealthy individuals (whether of normal weight, overweight, or obese) show an elevated risk of health complications [37]. It seems that in determining the risk of CVD and overall mortality, the distribution of body AT and other clinical parameters should be taken into account alongside the BMI value.” (page 4 line 155).

Conclusions:

  • Comment no 1.

The Authors write: “Understanding the underlying causes associated with various phenotypes of obesity may contribute to the development of a better therapeutic strategy, reducing the number of cases and complications and thereby enhancing the quality of life of the patients with an excessive body AT content.” – please discuss if are any examples of therapeutic strategies from pre- or clinical studies aiming at described adipokines/myokines in treatment of obesity and/or its complications.

Thank you for your comment. After literature searching, and the lack of unambiguous pre- or clinical studies describing the use of discussed adipokines/myokines as a direct strategy of obesity treatment, we decided to modify the conclusions. Instead previous sentence we added other one: “However, there are still not enough studies to determine the exact effect of these molecules on metabolism, and in particular on obesity phenotypes. Further research is thus warranted to assess the contribution of adipokine and myokine networks in obesity phenotypes, with the aim of identifying molecules of potential prognostic or diagnostic significance.” (page 15 line 671).

Minor revisions (whole manuscript):

  • Comment no 1.

The manuscript may benefit from the assistance of a native English speaker.

The revised version of the manuscript has been reviewed and edited by a native speaker of the English language. Certificate of English editing was added to the submission system.

Minor revisions (whole manuscript):

  • Comment no 1.

“Skeletal muscles, as the most abundant tissue in the human body, are responsible for generating movements and maintaining the balance of the body." – please add "in the normal-weight individuals.”

The sentence has been added (page 2 line 81).

We would like to add that the manuscript has been read carefully to correct and eliminate spelling, grammar and stylistic errors. We hope the revised version is now suitable for publication and look forward to hearing from you due to the course.

Yours sincerely,

Marta Pelczyńska                                                                                                    Corresponding Author

Reviewer 2 Report

Comments and Suggestions for Authors

The manuscript entitled: "The role of adipokines and myokines in the pathogenesis of 2 different obesity phenotypes – a new perspectives" explores a novel topic. Please find below some suggestions that will help readers understanding.

Abstract:

In the abstract, it would be helpful to add the difference/definition for different phenotypes of obesity. This way the abstract will be clear itself without relying on the main manuscript.

 Introduction

·       The introduction contains intriguing information, but it could benefit from reorganization. Specifically, it should address the following points in a clearer sequence: 1) Define adipokines and myokines. 2) Identify the tissues responsible for their production. 3) Explain their impact on obesity phenotypes.

·       Lines 58-59, Again, authors need to define obesity phenotypes in details and highlight the difference. It is missing in the introduction.

·       Line 58, The authors emphasize the significance of assessing the composition and distribution of adipose tissue (AT) in understanding obesity phenotypes, as individuals with similar BMI categories can have varying metabolic risks. It is important for them to provide further elaboration on how the composition and distribution of AT influence metabolic risk in this paragraph before transitioning to the skeletal muscles.

·       Line 73, Please provide examples of muscle-derived cytokines. In the current form looks very general.

·       Line 86-87, please rewrite the sentence.

·       Lines 115-116, NOW shows the higher risk compared to what?

·       Lines 155-57, how is nesfatin-1 related to these factors? Please make it more clear.

·       Line 202, please define “hyperphagia”,

·       Lines 244-247. Authors mention nesfatin-1 play a beneficial role in the development and maintenance of MHO. Even though MHO is a considered as a healthier phenotype of obesity, however it is still considered as obesity. Considering the beneficial effects of nesfatin-1 specially in increasing BAT, is it logical to say nesfatin-1 involves in “maintenance” of MHO? It seems misleading.

·       Line 293-295. In this section, please mention how is the relationship between Neuregulin 4 and obesity phenotypes. Is their any causal effects/relation?

·       Lines 315-318. So how to explain this contradiction that fasting and HFD show the same effects on myonectin.

·       Lines 324-332, It is not clear exactly how myonectin prevented from obesity in this study. Was it a correlation? Or authors addressed biological mechanisms?

·       Line 470, Seems to be a typo: “Another studies”

·       Line 478, Authors are right to mention it is difficult to attribute a specific role of decorin in obesity. However, to provide a better image of the condition it is better to categorize the studies in obese, non obese and different groups of obese subjects.

Author Response

RESPONSE TO REVIEWER 2:

We are grateful to you for taking your time to read our paper and for your comments. We have carefully reviewed the comments and have revised the manuscript accordingly. Our responses are given below in a point-by-point manner. All suggested corrections have been included in the modified version and were marked up using the “Track Changes” function in the text.

  • Comment no 1.

Abstract: In the abstract, it would be helpful to add the difference/definition for different phenotypes of obesity. This way the abstract will be clear itself without relying on the main manuscript.

Thank you for your comment. The definition of obesity phenotypes has been added to the abstract section: “Body mass index (BMI) is the key indicator in the diagnosis of obesity. However, in the case of the NWO and MONW phenotypes, the metabolic disturbances are present despite BMI being within the normal range. On the other hand, MHO subjects with elevated BMI values do not present metabolic abnormalities. The MUO phenotype involves both a high BMI value and an abnormal metabolic profile” (page 1 line 19).  

  • Comment no. 2

Introduction: The introduction contains intriguing information, but it could benefit from reorganization. Specifically, it should address the following points in a clearer sequence: 1) Define adipokines and myokines. 2) Identify the tissues responsible for their production. 3) Explain their impact on obesity phenotypes.

Thank you for your careful reading and comments. The introduction has been reorganized taking into account the Reviewer suggestions. The information addressed to the points 1-3 were described in separate, following paragraphs (page 1-3 line 42-103).

  • Comment no. 3

Lines 58-59, Again, authors need to define obesity phenotypes in details and highlight the difference. It is missing in the introduction.

Thank you for your comment. The description of obesity phenotypes has been added (page 2 line 70-76).

  • Comment no. 4

Line 58, The authors emphasize the significance of assessing the composition and distribution of adipose tissue (AT) in understanding obesity phenotypes, as individuals with similar BMI categories can have varying metabolic risks. It is important for them to provide further elaboration on how the composition and distribution of AT influence metabolic risk in this paragraph before transitioning to the skeletal muscles.

The description has been added in the third paragraph of Introduction (page 2 line 55-70).

  • Comment no 5.

Line 73, Please provide examples of muscle-derived cytokines. In the current form looks very general.

The examples have been given (page 2 line 89-91).

  • Comment no 6.

Line 86-87, please rewrite the sentence.

The sentence has been rewritten (page 3 line 110-112).

  • Comment no 7.

Lines 115-116, NOW shows the higher risk compared to what?

The sentence has been changed (page 4 line 137-141).

  • Comment no 8.

Lines 155-57, how is nesfatin-1 related to these factors? Please make it more clear.

As suggested, the sentence has been rewritten and explained (page 5 line 190-192).

  • Comment no 9.

Line 202, please define “hyperphagia”.

The term has been defined (page 5 line 237-238).

  • Comment no 10.

Lines 244-247. Authors mention nesfatin-1 play a beneficial role in the development and maintenance of MHO. Even though MHO is a considered as a healthier phenotype of obesity, however it is still considered as obesity. Considering the beneficial effects of nesfatin-1 specially in increasing BAT, is it logical to say nesfatin-1 involves in “maintenance” of MHO? It seems misleading.

Thank you for your comment. The sentence has been changed as suggested (page 6 line 279-282).

  • Comment no 11.

Line 293-295. In this section, please mention how is the relationship between Neuregulin 4 and obesity phenotypes. Is their any causal effects/relation?

Thank you for your valuable comment. The sentence has been changed and the relationship between neuregulin 4 and obesity has been described (page 9 line 330-334).

  • Comment no 12.

Lines 315-318. So how to explain this contradiction that fasting and HFD show the same effects on myonectin.

Thank you for the insight. The results has been explained and the paragraph has been rearranged (page 9 line 353-361).

  • Comment no 13.

Lines 324-332, It is not clear exactly how myonectin prevented from obesity in this study. Was it a correlation? Or authors addressed biological mechanisms?

The explanation has been added. Authors did not gave any correlation but indicated a potential relationship between myonectin, exercise and obesity, what was described in the revised version of the article (page 9-10 line 362-375).

  • Comment no 14.

Line 470, Seems to be a typo: “Another studies”

Yes, thank you for this comment. It was corrected (page 12 line 502).

  • Comment no 15.

Line 478, Authors are right to mention it is difficult to attribute a specific role of decorin in obesity. However, to provide a better image of the condition it is better to categorize the studies in obese, non obese and different groups of obese subjects.

The studies has been categorized in the summary of the decorin description (page 12 line 509-516).

We would like to add that the manuscript has been read carefully to correct and eliminate spelling, grammar and stylistic errors. Moreover, the article has been checked and corrected due to English language by a native speaker. We hope the revised version is now suitable for publication and we look forward to hearing from you due to the course.

Yours sincerely,

Marta Pelczyńska                                                                                                      Corresponding Author

Round 2

Reviewer 1 Report

Comments and Suggestions for Authors

I would like to express my gratitude for the possibility to re-review the paper entitled  "The role of adipokines and myokines in the pathogenesis of different obesity phenotypes – a new perspectives" by Marta Pelczyńska et al. Since the Authors addressed   manuscript concept and structure, I find the paper acceptable for publication in the  "Antioxidants" journal.